# Can the creation of new freshwater habitat demographically offset losses of Pacific salmon from chronic anthropogenic mortality?

**Pascale Gibeau**[1] *, **Michael J. Bradford**[2], **Wendy J. Palen**[1]

**1** Department of Biological Sciences, Earth to Ocean Research Group, Simon Fraser University, Burnaby, British Columbia, Canada, **2** Fisheries and Ocean Canada, Pacific Science Enterprise Centre, West Vancouver, British Columbia, Canada

* pgibeau@sfu.ca

**Data Availability Statement:** All relevant data are within the paper and its Supporting information files.

## Abstract

Over 1 billion USD are devoted annually to rehabilitating freshwater habitats to improve survival for the recovery of endangered salmon populations. Mitigation often requires the creation of new habitat (e.g. habitat offsetting) to compensate population losses from human activities, however offsetting schemes are rarely evaluated. Anadromous Pacific salmon are ecologically, culturally, and economically important in the US and Canada, and face numerous threats from degradation of freshwater habitats. Here we used a matrix population model of coho salmon (*Oncorhynchus kisutch*) to determine the amount of habitat offsetting needed to compensate mortality (2–20% per year) caused by a range of development activities. We simulated chronic mortality to three different life stages (egg, parr, smolt/adult), individually and simultaneously, to mimic impacts from development, and evaluated if the number of smolts produced from constructed side-channels demographically offset losses. We show that under ideal conditions, the typical size of a constructed side-channel in the Pacific Northwest (PNW) (3405 m$^2$) is sufficient to compensate for only relatively low levels of chronic mortality to either the parr or smolt/adult stages (2–7% per year), but populations do not recover if mortality is >10% per year. When we assumed lower productivity (e.g.; 25$^{th}$ percentile), we found that constructed channels would need to be 2.5–4.5 fold larger as compared to the typical size built in the PNW, respectively, to maintain population sizes. Moreover, when we imposed mortality to parr and smolt/adult stages simultaneously, we found that constructed side-channels would need to be between 1.8- and 2.3- fold larger that if the extra chronic mortality was imposed to one life stage only. We conclude that habitat offsetting has the potential to mitigate chronic mortality to early life stages, but that realistic assumptions about productivity of constructed side-channels and cumulative effects of anthropogenic disturbances on multiple life stages need to be considered.

**Funding:** NSERC Discovery Grant, the Gordon and Betty Moore, and Wilburforce Foundations to WJP.

**Competing interests:** The authors have declared that no competing interests exist.

## Introduction

Society spends billions of dollars annually to mitigate impacts of human activities on biotic communities and abiotic processes. The stakes of such ecological and economic trade-offs are high, as total investments in energy, water, and infrastructure development projects are expected to exceed $53 trillion (US) worldwide between 2010 and 2030 (OECD 2012 in [1]). Regulatory agencies often require that developers apply the mitigation hierarchy, which consists of avoidance, then minimization, and finally offsetting for the impact of projects on ecosystems and biodiversity [1, 2]. In many jurisdictions, a requirement for "No Net Loss" of biodiversity often results in the use of compensatory measures, such that losses from development are fully offset to maintain and stabilize population sizes after development [1]. For example, in Canada, the *Fisheries Act* (2015) includes a provision to employ offsetting when authorizing activities other that fishing that result in the death of fish.

Globally, most offsetting measures involve creating or restoring habitats, which is often done in "like-for-like" schemes, where offsetting attempts to replace areas of a given habitat lost to development by an equal or greater amount of the same habitat [2, 3]. When "like-for-like" offsetting is not feasible, managers may employ a different approach called "out-of-kind" offsetting, where improving conditions for a different population, or easing pressures from a different threat on the targeted population, may compensate the expected effects of development [1, 3]. For example, Barnthouse et al [4] describe a proposal to compensate fish mortality due to entrainment by water intake structures of a nuclear plant by removing a dam 50 km inland from the facility. Other researchers have proposed restoration actions in "out-of-kind" offsetting schemes to compensate for fishing mortality of American lobsters [5], and seabird bycatch [6]. However, to our knowledge, the potential for "out-of-kind" habitat offsetting to compensate ongoing chronic anthropogenic mortality at a population level has not been previously evaluated.

Here, we use a matrix population model of coho salmon (*Oncorhynchus kisutch*) in the Pacific Northwest (PNW) of North America to evaluate the efficacy of "out-of-kind" offsetting by estimating the amount of offsetting habitat that would be required to achieve "No Net Loss" of productivity for salmon populations affected by anthropogenic development activities that cause mortality. Coho salmon spawn in natal freshwater streams during late fall, and eggs incubate buried in the substrate over the winter. Fry emerge in the spring and quickly transform into juveniles called parr, which typically remain in freshwaters until their second spring, before migrating to the ocean as smolts. They finish their transformation to adulthood in the ocean, where they spend another 18 months before returning to freshwaters to spawn [7]. Most coho salmon populations experience density-dependent survival bottlenecks during their freshwater residency, most often in the spring immediately after fry emergence, since territorial fry compete for limited space and resources [8, 9]. Juvenile coho salmon rely on small streams as freshwater nursery habitat, which makes them vulnerable to development activities including the alteration of in-stream and adjacent terrestrial habitats from forestry practices [8–10], other land-use changes [11], as well as the creation of barriers to migration and the entrainment of fish in water intake structures of small and large hydropower dams (e.g. in the Columbia River system, [14]). The off-channel habitats used by coho salmon juveniles, like sloughs, side-channels, beaver ponds, or temporary to permanent floodplains, are thus often a target for restoration or offsetting actions [12, 13]. Coho salmon represent a good species for demonstrating the use of "out-of-kind" offsetting for chronic anthropogenic mortality as there is sufficient empirical information on population dynamics as well as data on the efficacy of offsetting or restoration measures.

We evaluated how much offsetting habitat (in $m^2$), in the form of constructed off-channel habitat, was required to offset chronic mortality on coho salmon and maintain overall

population size and productivity. Specifically, our objective was to assess how chronic anthropogenic mortality to three coho salmon life history stages (eggs, parr, smolt/adult), individually and simultaneously, influenced overall population dynamics, and thus, the effectiveness of habitat offsetting. Using threatened populations of coho salmon to develop a quantitative framework for balancing development losses and mitigation gains in "out-of-kind" schemes serves the important call for research into addressing the uncertainty in offset analysis [1]. We expected that the value of constructed offsetting habitats to the overall population dynamics of coho salmon would depend on which life history stage(s) experienced additional mortality. For example, we predicted that less offsetting would be needed when chronic mortality targeted individuals before they experienced density-dependent survival, or if individuals contributed to the main population by offsetting habitat did not also suffer the chronic mortality (e.g. when offsetting habitats are located downstream of intake structures of dams).

## Methods

### Scenarios

To evaluate the amount of habitat offsetting required to maintain productivity for coho salmon populations experiencing chronic mortality, we modelled the offsetting potential of constructed offsetting habitat for three life stages of coho salmon: egg, parr, and smolt/adult. We also evaluated how habitat offsetting requirements change when mortality occurs at multiple life stages simultaneously. We focused on constructed side-channels because of their importance for rearing and overwintering coho salmon juveniles [12, 14], and because they are commonly used to limit or reverse the decline of anadromous salmon in the PNW [15, 16].

We created scenarios that varied the chronic mortality imposed on each life stage as well as our assumptions about the number of smolts produced by offsetting habitats, to evaluate how chronic mortality could be offset by adding offsetting habitat. We used a 3-stage deterministic matrix population model with density-dependent survival in the fry to parr transition to assess impacts to population size. The first set of scenarios (Egg scenarios) imposed mortality on eggs prior to density-dependent survival. Egg scenarios encompass a wide range of activities known to cause anthropogenic mortality to the egg stage, including increased sediment deposition or scour of spawning areas following deforestation [8], thermal extremes [17], and dewatering of spawning areas below dams [18, 19]. In Parr scenarios, we represented chronic anthropogenic mortality caused by activities such as dam or powerplant entrainment [20, 21], flow fluctuations due to hydropower generation [22, 23] and degradation in water quality from forestry practices or urbanization [9, 24]. In Smolt/Adult scenarios, we imposed chronic mortality to represent impacts including smolt entrainment in, or passage over, dams [25, 26], or adult mortality during upstream migration due to warmer temperatures downstream of dams or powerplants [27]. In both the Parr and the Smolt/Adult scenarios, we assumed individuals experienced chronic anthropogenic mortality after the period of density-dependent survival (i.e., fry stage), but in the Smolt/Adult scenario, the smolts contributed by the constructed side-channels also suffered the extra chronic mortality experienced by the main population (e.g. in cases when offsetting habitats are constructed upstream of dams). We combined smolts and adults in the same stage in our models because data on survival rates between when smolts leave rivers and when spawning adults return were not available. Therefore, applying a chronic mortality rate at any point during that stage in our models was mathematically equivalent.

Precise estimates of mortality by classes of anthropogenic disturbance are rare, and relating stage-specific vital rates (e.g. survival) to degraded habitat conditions remains difficult [28]. Consequently, we simulated a broad range (2–20% per year) of chronic annual mortality to represent disturbances causing relatively small to large annual mortality (e.g. juvenile

entrainment in spillways or turbines of dams, [20]). We applied annual mortality rates to egg, parr, and smolt/adult stages, both separately and in combination, to evaluate the potential population-level consequences of chronic anthropogenic impacts, and the scope for offsetting habitats to ameliorate those effects.

## Deterministic matrix model

We created a deterministic 3-stage matrix model with a one-year time step to represent a typical 3-year coho salmon life-cycle [11, 15], and ran each simulation for 45 years to compare the final population size under baseline conditions to those for scenarios varying life stages affected, rates of chronic mortality, and sizes and productivity of offsetting habitat. The first stage ($F_{13}$) in our model thus includes adult fecundity ($P_{fem}*F_{eggs}$), egg ($\varphi_{em}$), and fry ($f_{(\varphi spr)}$) survival, while the second stage ($a_{21}$) includes the early months of adult survival ($\varphi_{oceY2}$) in the ocean (Table 1). The third stage ($a_{32}$) is made up of adult ocean survival as well as migration upstream into spawning habitats ($\varphi_{oceY3}$). We estimated the parameters used in our model by summarizing all data available in peer-reviewed publications from the PNW. We estimated $P_{fem}$ (proportion adult female) at 0.452 to reflect higher ocean mortality of adult females compared to males (mean of two creeks, [29, 30]) and $F_{eggs}$ (mean fecundity) at 2597 eggs per female (mean of 16 creeks, [31, 32]). We assumed a survival to emergence, $\varphi_{em}$, of 0.223 based on studies from five creeks of the Pacific Northwest [33, 34]. Finally, we computed an overall

**Table 1. Deterministic matrix model and scenarios per life stage.**

$$\begin{bmatrix} 0 & 0 & F_{13} \\ a_{21} & 0 & 0 \\ 0 & a_{32} & 0 \end{bmatrix} \times \begin{pmatrix} N_p \\ N_s + N_{s_compH} \\ N_a - N_{a_compH} \end{pmatrix} = \begin{pmatrix} N_p \\ N_s \\ N_a \end{pmatrix}$$

| Transition rate | aij | Parameter equations |
|---|---|---|
| *Egg scenarios* | | |
| Eggs to parr | $F_{13}$ | $P_{fem} * (F_{eggs} * \boldsymbol{\varphi_{dist}}) * \varphi_{em} * f(\varphi_{spr})$ |
| Parr to smolt | $a_{21}$ | $\varphi_{oceY2}$ |
| Smolt to adult | $a_{32}$ | $\varphi_{oceY3}$ |
| *Parr scenarios (chronic mortality before offsetting)* | | |
| Eggs to parr | $F_{13}$ | $P_{fem} * F_{eggs} * \varphi_{em} * f(\varphi_{spr}) * \boldsymbol{\varphi_{dist}}$ |
| Parr to smolt | $a_{21}$ | $\varphi_{oceY2}$ |
| Smolt to adult | $a_{32}$ | $\varphi_{oceY3}$ |
| *Smolt/Adult scenarios (chronic mortality after offsetting)* | | |
| Eggs to parr | $F_{13}$ | $P_{fem} * F_{eggs} * \varphi_{em} * f(\varphi_{spr})$ |
| Parr to smolt | $a_{21}$ | $\varphi_{oceY2}$ |
| Smolt to adult | $a_{32}$ | $\boldsymbol{\varphi_{dist}} * \varphi_{oceY3}$ |

$\boldsymbol{\varphi_{dist}}$ was highlighted in bold to emphasize that its location is the only one that changes across scenarios.

$P_{fem}$ = proportion of females spawning (0.452); $F_{eggs}$ = # of eggs per female (2597); $\varphi_{em}$ = survival from hatching to emergence (0.223); $f(\varphi_{f\_spr})$ = density-dependent survival of fry; $\varphi_{oceY2}$ = survival of smolt through 1st summer and fall in ocean (0.392); $\varphi_{oceY3}$ = survival of adults in ocean (Year 3, 0.154); $\varphi_{dist}$ = survival after chronic anthropogenic disturbance (2, 5, 7, 10, 15, or 20%); $N_a$ = Number of adults; $N_{a\_compH}$ = Minimum number of adults needed to seed the offsetting habitat; $N_p$ = Number of parr; $N_s$ = Number of smolts; $N_{s\_compH}$ = Number of smolts contributed by the offsetting habitat.

ocean survival of 6.04% based on the geometric mean of recent data from three wild coho populations from the Pacific Northwest (prior to harvesting, [35, 36]), which reflects the high ocean mortality suffered by coho populations in recent years. We then split the ocean survival rate for each brood year between the six months of ocean survival in Year 2 ($\varphi_{oceY2}$) and 12 months in the ocean of Year 3 ($\varphi_{oceY3}$).

We assumed survival from fry to smolt was density-dependent with a bottleneck occurring immediately after fry emergence in the spring, since territorial fry compete for limited resources [37–39]. The density-dependent relationship was modelled with a Beverton-Holt function [40]:

$$\text{f}_{(\varphi\text{spr})} = \frac{\alpha \ \times \ \text{D}_{\text{fry}(t)}}{1 + \left[ \left(\frac{\alpha}{k}\right) \times \text{D}_{\text{fry}(t)} \right]} \tag{1}$$

Where $\alpha$ is the number of smolts per fry at the origin, $\text{D}_{\text{fry}(t)}$ is the number of fry at emergence (i.e. $P_{fem} * F_{eggs} * \varphi_{em}$), and $k$ is the carrying capacity for smolts in the stream (smolts per km). We modeled freshwater density-dependent survival after fry emergence by assuming $\alpha = 0.5$ which is the average from 10 coho salmon populations in the PNW [33, 35]. We fixed the carrying capacity for smolts ($k$) at 15,318, corresponding to the mean $k$ from field estimates (i.e. 1702 smolts/km, [35, 41]), multiplied by 9 km, the average length of the spawning or rearing reaches of 10 streams.

We ran the deterministic model in the absence of added anthropogenic mortality with an initial population size of 500 adults to calculate the stable final population sizes under baseline conditions after 45 years (808 adults, 13,379 parr, and 5249 smolts). We used the stable final population sizes at equilibrium as the starting population vector for all subsequent simulations.

We assumed offsetting side channels were fully functional immediately after construction, produced smolts at the maximum capacity each year ($k$), and the capacity to produce smolts remained constant over the length of the simulations. Spawning adults were first allocated to constructed side-channels until fully seeded ($N_{a\_compH}$, varied with the size of offsetting simulated in each scenario), after which any remaining spawning adults were allocated to the main channel. Assuming that the offsetting habitat is seeded at $k$ at all time effectively assumes that increasing size of offsetting habitat directly increases overall $k$ of the system. We computed $N_{a\_compH}$ by using average vital rates,

$$Na_{compH} = \frac{N_{s_c ompH}}{(P_{fem} * \ F_{eggs} * \ \phi_{em} * \phi_{f\_s})}$$

where $N_{s\_compH}$ is the number of smolts contributed by the offsetting habitat (varied with the size of side-channels in each scenario) and $\phi_{f\_s}$ is the average survival rate from fry to smolts in the absence of density dependence (0.075).

We calculated how many smolts would need to be produced from constructed offsetting habitats in order to maintain population productivity (i.e. achieve offsetting equivalency), defined as population abundance returning to baseline (pre-impact) levels within 45 years. To do so, we ran simulations for each chronic mortality rate and impacted life stage over a range of offsetting habitat sizes constructed for coho salmon in the PNW (Fig 1, n = 27 sites, [12–14]), and evaluated if offsetting equivalency was achieved in 45 years. In Parr scenarios, we assume that the offsetting habitats were constructed downstream of the source of mortality so that smolts produced in offsetting habitats were not exposed to the anthropogenic mortality. In comparison, in Smolt/Adult scenarios, chronic anthropogenic mortality impacted smolts or adults from both the offsetting habitat and main population. Finally, we assessed the additional

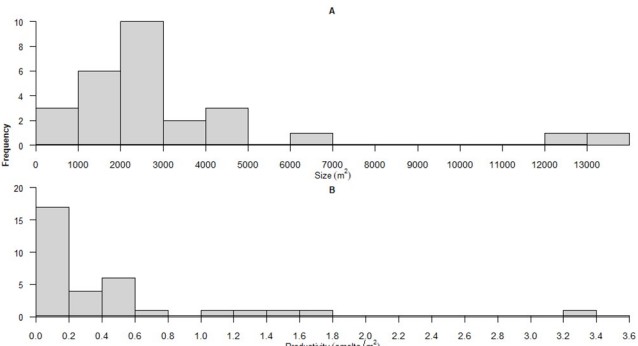

**Fig 1. Distribution of (a) the size (m²) of 27 side-channel habitats built in the PNW from [12–14], and (b) the productivity of 33 side-channel habitats (# of smolts per m²) built in the PNW from [14, 54].**

offsetting requirements if chronic mortality occurred during both parr and smolt/adults stages. To do so, we ran simulations with $\phi_{dist}$ applied to parr in the second year of the life cycle (*a21*), and to smolt/adults in the third year (*a32*).

## Offsetting channels

Finally, our simulations also explored how assumptions about the productive capacity (i.e. quality, expressed as # of smolts produced per m²) of offsetting habitats influenced the size of habitats needed to achieve offsetting equivalency. In the baseline scenario we assumed a mean smolt production in side-channels of 0.47 smolts per m², corresponding to the mean production estimated from 33 constructed side-channels in the PNW [14, 42] (Fig 1b). We also relaxed these assumptions and used the 25th (0.1 smolt / m²), 50th (0.18 smolts / m²), and 75th (0.54 smolts /m²) percentiles of smolt production as additional scenarios. All analyses were performed in program R (version 3.5.1, R Core Team 2013 [43]) and an example of model R code used for simulations is provided in S1 File.

## Elasticity analyses

We performed a simulation-based elasticity analysis to assess how proportional changes in population growth rate changed relative to proportional changes in the individual vital rates included in the deterministic matrix model without density dependence [44]. We created 10,000 matrices with vital rates drawn at random from uniform distributions between the upper and lower 95% confidence interval for each vital rate, and calculated the deterministic growth rate (λ) [45]. Density-dependent fry survival ($f(\varphi_{spr})$) was replaced by deterministic fry survival in year 1 ($\varphi_{fryY1}$) and year 2 ($\varphi_{fryY2}$), that averaged 7.5%, the mean fry-to-smolt survival across the 10 creeks used to derive the Beverton-Holt density-dependent function [33, 35]. We used a general linear model with Gaussian distribution to decompose the variation in lambda, in which the standardized regression coefficients associated with each vital rate in the model estimate the proportional contribution of each vital rate to the variation observed in lambda (i.e. elasticity, [44]).

## Results

The population-level effect of chronic anthropogenic mortality depended on whether the chronic mortality occurred before or after the stage when density-dependent mortality occurred (e.g. fry stage). When we modeled egg mortality without offsetting, the impact on

final population sizes was modest, with no more than a 3% reduction from the baseline population size. In comparison, when chronic mortality was imposed on parr or smolts/adults, i.e. after the period of freshwater density dependence, the final population size declined as chronic mortality increased (Fig 2).

Simulated populations that experienced chronic anthropogenic mortality benefitted from the addition of smolts from constructed side-channels, regardless of the life stage affected (Fig 2). We found that final population sizes increased linearly with the size of constructed side-channels until, and beyond, achieving offsetting equivalency (i.e. reaching adult abundances similar to baseline levels). However, we found that the importance of side-channel offsetting habitat varied depending on the timing of mortality in the coho salmon life cycle. For example, a relatively small constructed side-channel was sufficient to achieve offsetting equivalency when mortality impacted the egg stage, but the size of side-channels required to offset mortality occurring on the later life stages was much greater. For example, a small side-channel of approximately 1000 m$^2$ (i.e., smaller than the 10$^{th}$ percentile of side-channels built in the PNW) could compensate for up to 20% chronic annual mortality to the egg stage, but only 2% chronic annual mortality to parr or smolts/adults. Generally, more offsetting habitat was needed to achieve offsetting equivalency when smolts/adults were affected compared to parr, especially if the intensity of chronic mortality was greater than 5% annually.

When chronic mortality was applied to both parr and smolt/adult life stages, the size of offsetting habitat needed to reach offsetting equivalency increased with the magnitude of chronic mortality (Fig 3). The cumulative, or combined, effect of extra mortality on parr and smolt/

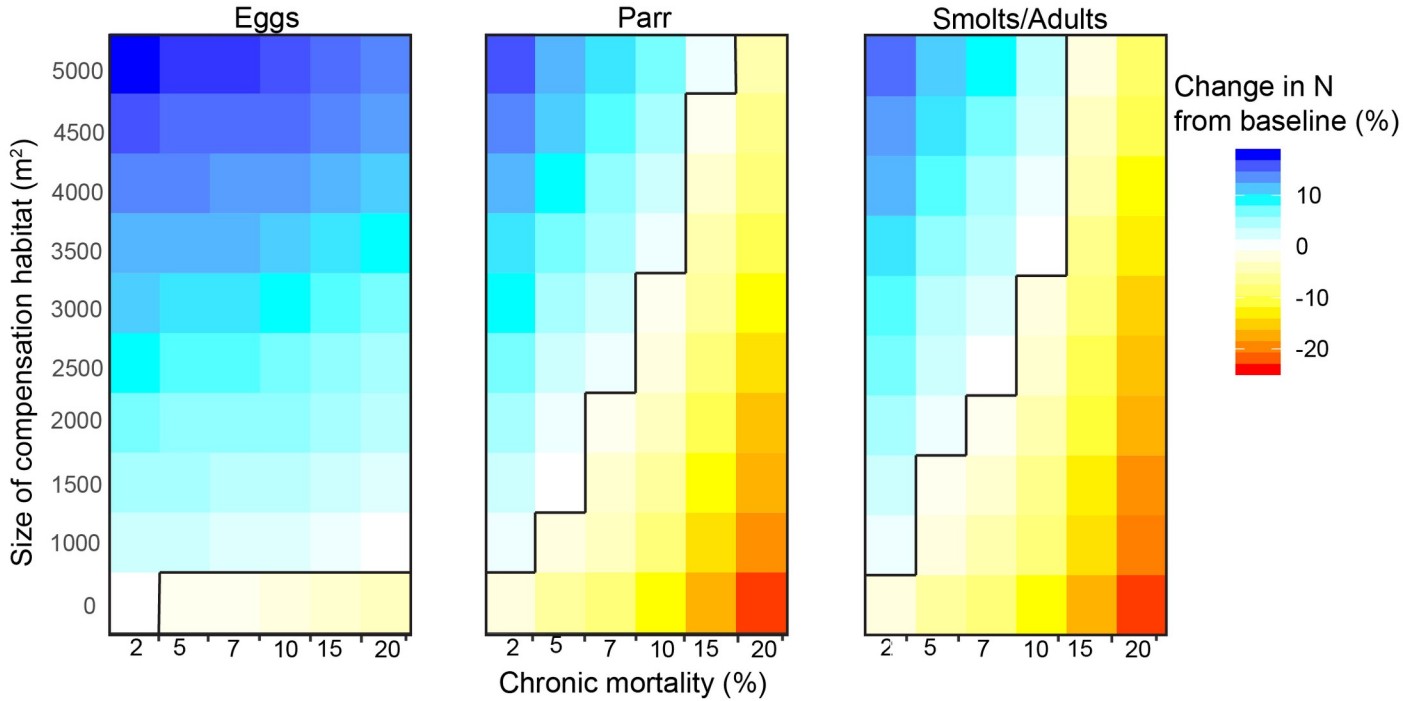

**Fig 2. Change in final number of adults for impacted populations compared to baseline abundances (%, colors) over a range of sizes of constructed side-channels (y-axis), annual chronic anthropogenic mortality (x-axis), and life stage (egg, parr, smolt/adult) affected by the chronic mortality (panels), assuming mean productivity in constructed side-channels.** 0 (white cells) indicates no change in final abundances of impacted populations compared to baseline, while positive values (cool shades) mean offsetting increased the final number of adults in impacted populations and negative values (warm shades) mean final size of impacted populations decreased despite the amount of offsetting habitat added. The black lines indicate offsetting equivalency for the combination of offsetting habitat sizes and chronic mortality.

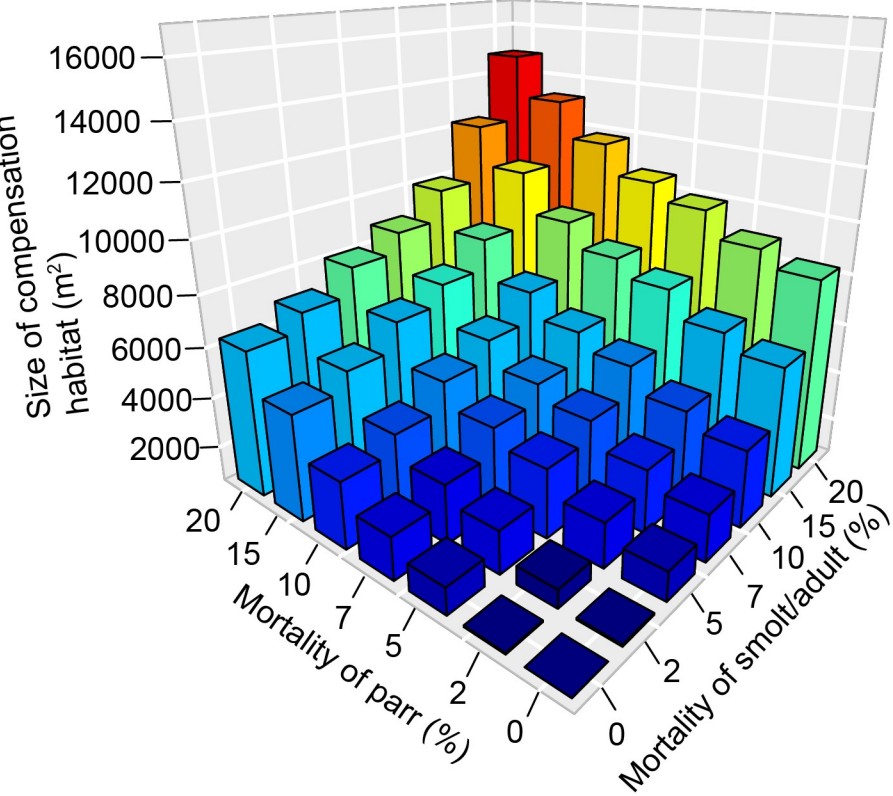

**Fig 3. Range in size of offsetting habitat (m$^2$) (z-axis) required to effectively offset annual chronic mortality ranging from 0 to 20% when chronic mortality is applied cumulatively to both parr (x-axis) and smolt/adult (y-axis) life stages.** We assumed mean productivity in constructed side-channels (0.47 smolts per m$^2$).

adult stages was marginally additive, but became multiplicative when anthropogenic mortality to smolt/adults was greater than 7%, and mortality to parr above 10%. For example, when an additional 20% mortality was applied to both parr and smolt/adults stages annually, the size of constructed side-channels needed to achieve equivalency increased by 1.8 to 2.3 times (from 6,336 m$^2$ or 8259 m$^2$, respectively, to 14,867 m$^2$) compared to when the same chronic mortality was applied to either the parr or smolt/adult stage separately.

## Productivity of offsetting habitats

Our results suggest that assumptions regarding the productivity of offsetting habitats (i.e., the number of smolts produced by the constructed side-channels) have a large effect on our conclusions regarding the size of offsetting needed to achieve offsetting equivalency. Based on the 25$^{th}$ percentile of productivity, the size of side-channels required to achieve offsetting equivalency was 4.5 times greater than if we assumed mean quality (Fig 4).

## Elasticity analyses

Our simulation-based elasticity analyses assessed how population growth rates changed relative to proportional changes in vital rates and indicated that survival of fry to smolt ($\phi_{f\_s}$ elasticity of 0.15) and adult ocean survival (elasticity of 0.16) had the largest influence on deterministic population growth rates. These two rates were three times more important than fecundity (*Feggs*) and

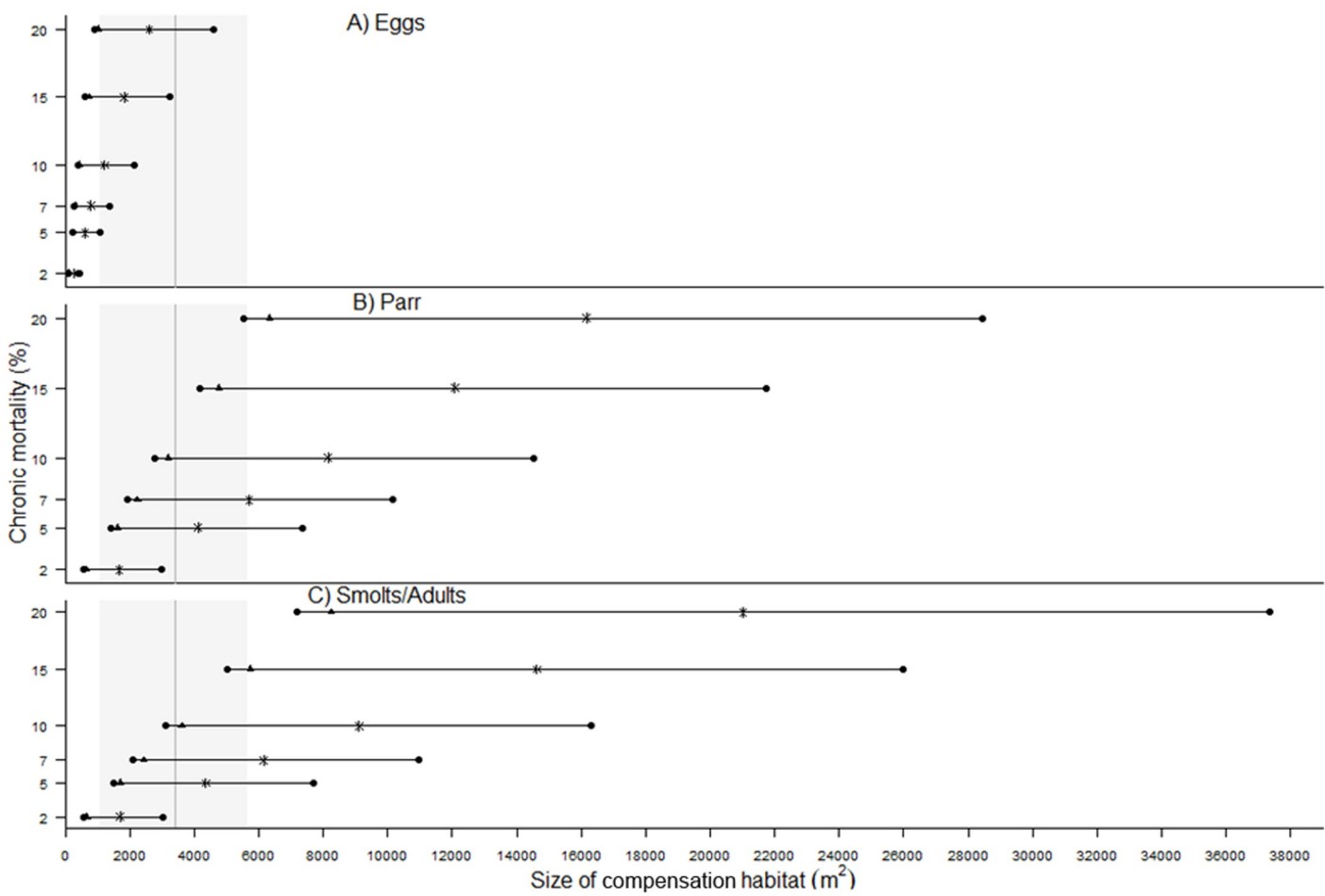

**Fig 4. Range in sizes of offsetting habitat (m²) required to effectively offset annual chronic mortality ranging from 2 to 20% affecting, a) Egg, b) Parr, and c) Smolt/Adult life stages, assuming varying productivity (# of smolts produced per m²) for the constructed side-channels.** Horizontal black lines indicate the 25th to 75th percentiles of productivity of offsetting habitat, black triangles represent mean number of smolts contributed, and stars represent median number of smolts contributed by offsetting habitat (based on data from 33 sites, [14, 54]). The grey box represents the mean (vertical line) and 10th to 90th percentiles (shaded) of sizes of offsetting habitats built in the PNW (n = 27 sites, data from [12–14]).

the survival of fry at emergence (elasticity of 0.051, 0.049, respectively), while the proportion of females (*Pfem*) had very little influence on population growth (elasticity of 0.0007).

## Discussion

We used a matrix population model to compare how mortality at one life stage can or cannot be offset by adding production of the same or a different life stage from offsetting habitat, in an equivalency analysis using an "out-of-kind" scheme to meet requirement for No Net Loss of population abundance. Our models suggest that the average size of constructed side-channels typically built in the PNW could compensate for chronic mortality of up to 20% annually if it affected the egg stage, but only up to 7% if it affected parr, smolts, or adults. Averaged-size side-channels would not be sufficient if added annual mortality was greater than 10% for parr or smolt/adult stages, greater than 5% if both parr and smolt/adult stages were affected, or if the productivity of side-channels is lower than the average values used in our baseline scenario. Constructed side-channels have a wide range of productivities, measured as the number of coho salmon smolts produced per m², because of differences in design and site-specific

considerations [14, 42]. Thus, a more precautionary approach to building side-channels for coho salmon would assume less than ideal productivity in offsetting habitats. If we assumed lower productivity, side-channels would need to be bigger than those required when we assumed mean productivity (*sensu* [14, 42]). Additionally, our results suggest that offsetting habitats are more effective at mitigating chronic mortality if they are built downstream of where disturbances occur (i.e., as in the parr scenarios), such that smolts they produce are not affected by chronic mortality experienced by the population in the main channel (which happens in the smolt/adult scenarios). Overall, if chronic mortality also affects the smolts contributed by the offsetting habitats (e.g. in cases of smolts entrainment into downstream dams or increased ocean mortality), achieving offsetting equivalency will require more offsetting habitat to be built in freshwater.

As our modelling of an "out-of-kind" offsetting scheme highlights, designing effective side-channels for offsetting is complex. Our results emphasize the importance of considering variation in productivity when deciding on the optimal size of offsetting habitat needed. However, the size of constructed side-channels may also impact quality of the offsetting habitat. For example, studies have noted a decline in smolt density with increasing side-channel sizes, suggesting that smaller side-channels may be more productive than large side-channels [14, 46]. Building numerous, but smaller, side-channels may also be more technically and economically feasible than fewer large ones. In our models, we assumed that larger offsetting habitats could be equally as productive as smaller habitats, but if productivity declines with increasing size or through time due to degradation, our results may underestimate the sizes needed to maintain population equivalency. In addition, we simulated constructed side-channels to always be fully seeded (i.e. used at maximum density by spawning adults), and this conservative assumption, if incorrect, may further underestimate the size of habitat needed to offset chronic mortality.

Adding to the complexity of designing constructed side-channels of adequate size and quality, the size and the number of side-channels are frequently chosen based on availability of land and costs of restoration rather than based on ecological bottlenecks or the potential for success [14]. Such an opportunistic selection of offsetting sites is stated as one reason why "No Net Loss" of productivity is not often achieved in practice, since offsetting habitat that is not structurally functional may have little effect on salmon populations [47]. Our results suggest that intentionally siting side-channels downstream of where anthropogenic impacts occur would be more effective at compensating for chronic mortality. However, downstream reaches of streams and rivers are often less stable geomorphologically [48, 49], and the substantial investment required to build offsetting habitats in low-elevation floodplains could be lost if, or when, natural large flow events occur.

Other challenges to building effective side-channels habitats include the need for connectivity between the offsetting habitats and main channel, which is crucial to ensure the success of mitigation [50, 51]. For example, schooling of juveniles and limited migration may lower the carrying capacity of constructed side-channels by limiting the number of fry that move in from spawning or rearing grounds [52]. Salmon also need dynamic and diverse freshwater habitats to thrive, and offsetting projects will be more effective if they consider natural evolutionary processes needed for the health and resilience of the species [53–55], especially considering the added challenges posed by accelerating climate change. For example, improving temperature and flow regimes through riparian restoration and increased food availability may improve the tolerance of salmon populations to warmer water temperatures induced by climate change [51]. Our results support the conclusion that cumulative impacts from multiple sources of anthropogenic mortality on more than one life-stage simultaneously can have compounding (i.e. multiplicative) effects on population dynamics, and increase the need for mitigation and offsetting activities. As such, our simulations of only a single life stage experiencing

chronic mortality are likely to be underestimates of the minimum sizes of constructed side-channels needed to maintain population sizes in the face of multiple, overlapping sources of anthropogenic mortality (e.g. [9]). Finally, the sound application of our modeling approach requires realistic estimates of mortality resulting from development projects. Offsetting requirements are usually determined during project design phases, and the mortality associated with anthropogenic activities are usually estimated from other sources of knowledge and coupled with site-specific information about populations that may be affected by the project. Commonly, estimated or predicted mortality rates are highly uncertain. However, that uncertainty can be incorporated by our modeling approach to evaluate its effects on offsetting requirements.

At the population level, our results highlight that the potential for effective mitigation of added chronic mortality depends both on understanding the influence of density-dependent survival in affected populations, as well as on the unequal contributions of different life stages to the overall population dynamics. For example, natural density-dependent bottlenecks may compensate for some mortality caused by anthropogenic disturbances if it occurs prior to periods of negative density dependent survival [56], and influence the need for offsetting habitat. Our results showing only modest population-level impacts of additional egg mortality illustrate the low elasticity of the egg stage to affect overall population dynamics, as well as the potential for density-dependent survival to compensate for some anthropogenic mortality. Independently from the issue of density dependence, our elasticity analysis also highlights the large influence that variation in fry-to-smolt survival (elasticity of 0.15) has on population growth, which further emphasizes the importance of survival at that stage for salmon conservation efforts. However, it is risky to rely on natural density dependence to compensate for chronic anthropogenic mortality as it remains very challenging to detect if, when and how strong density-dependent survival bottlenecks occur in freshwater for specific populations [52, 57, 58]. Moreover, density-dependent survival may only compensate for additional mortality when population sizes are large (e.g. approaching carrying capacity, [59]), which may be uncommon for many anadromous salmon populations currently impacted by anthropogenic disturbances [60], unless such disturbances also lower freshwater carrying capacity [61]. Finally, other forms of density dependence may enhance the efficacy of offsetting habitat in mitigating the effects of disturbances. For example, density-dependent migration, whereby individuals in areas with high densities move to areas with lower densities, allows the offsetting habitat to be more effectively populated by encouraging more individuals to colonize it [61]. Overall, given the difficulty in adequately assessing the presence, timing, and strength of density-dependent survival in freshwater, long-term studies of population dynamics of local populations appear crucial to adequately design effective offsetting habitats.

## Conclusion

Our results suggest that offsetting habitats built for coho salmon in freshwater have the potential to mitigate some chronic and ongoing mortality. However, our results also indicate that achieving offsetting equivalency may require creating a much larger amount of side-channel habitat than is typically constructed in the PNW [14, 42], especially when we relaxed assumptions about ideal smolt productivity in constructed side-channels, or considered cumulative impacts to multiple life stages. Moreover, our results illustrate how life-cycle population models can be a powerful tool to examine the efficacy of restoration efforts targeted to different life stages for overall population dynamics. We show that matrix population models can be used to quantitatively estimate the uncertainty in "out-of-kind" offset analysis, though their utility can be limited by the availability of site-specific information on population productivity and

the magnitude of anthropogenic mortality. Protecting and restoring threatened salmon populations in the PNW requires a critical evaluation of the efficacy of policies and practices of industry and regulators. Overall, the importance of side-channels for juvenile coho salmon and the large influence of fry-to-smolt survival rates on overall population dynamics suggested by our elasticity analysis highlight the value of building side-channels as offsetting habitat for coho salmon, despite the complexity and challenges involved.

## Supporting information

**S1 File.**
(DOCX)

## Acknowledgments

We thank K. Wilson, D.A Greenberg, A. Cantin, R. Murray, and J.W. Moore for help and comments that greatly improved the manuscript.

## Author Contributions

**Conceptualization:** Pascale Gibeau, Michael J. Bradford, Wendy J. Palen.

**Data curation:** Pascale Gibeau, Michael J. Bradford.

**Formal analysis:** Pascale Gibeau, Michael J. Bradford, Wendy J. Palen.

**Funding acquisition:** Wendy J. Palen.

**Investigation:** Pascale Gibeau.

**Methodology:** Pascale Gibeau, Michael J. Bradford, Wendy J. Palen.

**Project administration:** Pascale Gibeau.

**Resources:** Pascale Gibeau, Wendy J. Palen.

**Software:** Pascale Gibeau.

**Supervision:** Michael J. Bradford, Wendy J. Palen.

**Validation:** Pascale Gibeau, Michael J. Bradford.

**Visualization:** Pascale Gibeau, Michael J. Bradford, Wendy J. Palen.

**Writing – original draft:** Pascale Gibeau, Wendy J. Palen.

**Writing – review & editing:** Michael J. Bradford.

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
