## [Decision Letter · Decision Letter 0]

23 Sep 2020

PONE-D-20-21844

Can the creation of new freshwater habitat demographically offset losses of Pacific salmon from chronic anthropogenic mortality?

PLOS ONE

Dear Dr. Gibeau,

Thank you for submitting your manuscript to PLOS ONE. After careful consideration, we feel that it has merit but does not fully meet PLOS ONE’s publication criteria as it currently stands. Therefore, we invite you to submit a revised version of the manuscript that addresses the points raised during the review process.

We look forward to receiving your revised manuscript.

Kind regards,

Rachel A Hovel

Academic Editor

PLOS ONE

Journal Requirements:

2.We note that you have indicated that data from this study are available upon request. PLOS only allows data to be available upon request if there are legal or ethical restrictions on sharing data publicly. For more information on unacceptable data access restrictions, please see http://journals.plos.org/plosone/s/data-availability#loc-unacceptable-data-access-restrictions.

4. Please ensure that you refer to Figure 4 in your text as, if accepted, production will need this reference to link the reader to the figure.

Additional Editor Comments (if provided):

Thank you for your submission and for your patience as we worked through difficulty in securing a two suitable reviewers. I will be pleased to receive and reconsider a revised version of this manuscript.

Both reviewers saw substantial merit in this manuscript and its contributions, and comments focused largely on presentation and detail. In particular, please pay attention to the following points in addressing reviewer comments:

-ensure that the introduction and discussion do not over-reach the scope of the analysis

-provide additional detail on parameters and rationale for different models

Reviewers' comments:

Reviewer's Responses to Questions

**Comments to the Author**

1. Is the manuscript technically sound, and do the data support the conclusions?

Reviewer #1: Yes

Reviewer #2: Yes

2. Has the statistical analysis been performed appropriately and rigorously? 

Reviewer #1: Yes

Reviewer #2: Yes

3. Have the authors made all data underlying the findings in their manuscript fully available?

Reviewer #1: Yes

Reviewer #2: No

4. Is the manuscript presented in an intelligible fashion and written in standard English?

Reviewer #1: Yes

Reviewer #2: Yes

5. Review Comments to the Author

Reviewer #1: Review of “Can the creation of new freshwater habitat demographically offset losses of Pacific salmon from chronic anthropogenic mortality?”

Gibeau et al.

The authors tackle a very complex issue within the sphere of freshwater restoration and mitigation for anthropogenic impacts to these habitats. Development of these types of models are beneficial to managers who need data-driven tools to justify the size or scope of mitigation actions required to offset adverse impacts of human activities.

The scope of this analysis is more limited than expressed in the introduction and discussion. For this tool to be useful to managers, model inputs would be needed from local impacted populations. As is, coho productivity and compensation habitat size inputs are from a number of streams in a relatively small geographic area-SW British Columbia and NW Washington-nothing evident from larger systems like the Fraser or Columbia Rivers. However, the authors do an excellent job of demonstrating the potential of such an analysis if the inputs are available. The results are clearly communicated along with a good discussion of how the results may differ given variation in productivity estimates or mortality impacting more than one life stage. The greatest weakness of this analysis is that it relies on knowing the magnitude of added mortality due to human impacts, which is difficult to predict for an impact that has yet to happen. The authors acknowledge this shortcoming in the closing sentences of the discussion but do not elaborate further. A deeper discussion of the limitations of the model should be included.

The authors state that data are available upon request, however, more information on the geographic nature of the data used in the model would be useful to the reader.

Overall, the paper is very well written and will certainly contribute to the literature on quantifying and compensating for human impacts to salmon species.

A few specific comments:

References to figures need to be double-checked.

There is a large section where the caption for figure 2 has been incorporated into the text (lines 237-247).

There is a missing citation on line 210.

Reviewer #2: My review is attached as a word document.

6. PLOS authors have the option to publish the peer review history of their article (what does this mean?). If published, this will include your full peer review and any attached files.

Reviewer #1: No

Reviewer #2: No

---

## [Author Response · Author response to Decision Letter 0]

7 Nov 2020

Dear Dr. Hovel,

We are grateful for the opportunity to revise our manuscript in response to the helpful comments made by the two reviewers and yourself. We edited and expanded our manuscript to include almost all the suggestions made by the two reviewers. We addressed below all the comments provided by the two reviewers in a numbered, point-by-point response (shown in blue and italic), including indications as to where in the manuscript specific changes have been made. Specifically, we heavily edited the whole introduction and revised the discussion of the manuscript to refine the scope of the analysis (e.g. 283-285, 313-319). We also provided more detail on parameters and rationale for our simulations (e.g. lines 106-114, 131-141, 163-169). 

Additionally, we ensured that our manuscript meets PLOS ONE’s style requirements, included affiliation for all authors, and updated the references to all figures. We also would like to clarify that all the data needed to run our model and perform our simulations are indeed included in our manuscript (specifically in Methods and in Table 1) and apologize for the initial confusion surrounding data access. We added text to the methods regarding how we derived each model parameter, and include references to the published data sources for each (lines 131-141). We also now include in the Supplemental Material a simplified example of the R code used to run our model to ensure the replicability of our study.

We believe that these changes have improved our manuscript. We consider that this study remains of great interest by building on population dynamics in order to evaluate a real-world environmental problem with direct application to sound management of populations and habitats impacted by development. We hope that you will find that it meets the standard of scholarship and writing for PlosOne, and is suitable for publication. 

Sincerely,

Pascale Gibeau, PhD

Earth to Ocean Research Group

Simon Fraser University

---

## [Decision Letter · Decision Letter 1]

4 Dec 2020

Can the creation of new freshwater habitat demographically offset losses of Pacific salmon from chronic anthropogenic mortality?

PONE-D-20-21844R1

Dear Dr. Gibeau,

We’re pleased to inform you that your manuscript has been judged scientifically suitable for publication and will be formally accepted for publication once it meets all outstanding technical requirements. Thank you for your substantive revisions, which have improved clarity and readability. 

Kind regards,

Rachel A Hovel

Academic Editor

PLOS ONE

Additional Editor Comments (optional):

I appreciate your thorough revisions, and find this manuscript to now be ready for publication. Thank you for your attention to the reviewer comments, and for submitting this interesting paper. 

Reviewers' comments:

Reviewer's Responses to Questions

**Comments to the Author**

1. If the authors have adequately addressed your comments raised in a previous round of review and you feel that this manuscript is now acceptable for publication, you may indicate that here to bypass the “Comments to the Author” section, enter your conflict of interest statement in the “Confidential to Editor” section, and submit your "Accept" recommendation.

Reviewer #2: All comments have been addressed

2. Is the manuscript technically sound, and do the data support the conclusions?

Reviewer #2: Yes

3. Has the statistical analysis been performed appropriately and rigorously? 

Reviewer #2: Yes

4. Have the authors made all data underlying the findings in their manuscript fully available?

Reviewer #2: Yes

5. Is the manuscript presented in an intelligible fashion and written in standard English?

Reviewer #2: Yes

6. Review Comments to the Author

Reviewer #2: Reviewer Comments: Can the creation of new freshwater habitat demographically offset losses of Pacific salmon from chronic anthropogenic mortality? (PONE-D-20-21844_R1)

The authors did an excellent job providing thoughtful responses to mine and the other reviewer’s comments and I find that the revised manuscript is much improved. I especially appreciate how the authors elaborated on their use of coho salmon as an example species in the introduction and discussion sections and provide a more detailed explanation of their use of the term “productivity.” I laud their decision to include the model code as supplementary material. I still feel that the conclusion section could focus more on the big picture. I like to see the conclusion section (and/or the last paragraph of the discussion section) framed in terms that are as broad as those laid out in the first paragraph of the introduction section, but I won’t press this further as it does not detract from the overall message of the manuscript. Thanks to the authors for an interesting and informative read.

7. PLOS authors have the option to publish the peer review history of their article (what does this mean?). If published, this will include your full peer review and any attached files.

Reviewer #2: No

---

## [Editor Report · Acceptance letter]

9 Dec 2020

PONE-D-20-21844R1 

Can the creation of new freshwater habitat demographically offset losses of Pacific salmon from chronic anthropogenic mortality? 

Dear Dr. Gibeau:

I'm pleased to inform you that your manuscript has been deemed suitable for publication in PLOS ONE. Congratulations! Your manuscript is now with our production department. 

Kind regards, 

on behalf of

Dr. Rachel A Hovel 

Academic Editor

PLOS ONE